# Exploring the Mystery of Influential Data for Mathematical Reasoning

**Xinzhe Ni** [*]
Institute of Data and Information
Tsinghua University
Shenzhen, China
Microsoft
Beijing, China
nxz22@mails.tsinghua.edu.cn

**Yeyun Gong** [†]
Microsoft
Beijing, China
yegong@microsoft.com

**Zhibin Gou**
Institute of Data and Information
Tsinghua University
Shenzhen, China
Microsoft
Beijing, China
gzb22@mails.tsinghua.edu.cn

**Yelong Shen**
Microsoft
Beijing, China
yeshe@microsoft.com

**Yujiu Yang** [†]
Institute of Data and Information
Tsinghua University
Shenzhen, China
yang.yujiu@sz.tsinghua.edu.cn

**Nan Duan & Weizhu Chen**
Microsoft
Beijing, China
{nanduan, wzchen}@microsoft.com

## Abstract

Selecting influential data for fine-tuning on downstream tasks is a key factor for both performance and computation efficiency. Recent works have shown that training with only limited data can show a superior performance on general tasks. However, the feasibility on mathematical reasoning tasks has not been validated. To go further, there exist two open questions for mathematical reasoning: how to select influential data and what is an influential data composition. For the former one, we propose a **Q**uality-**a**ware **D**iverse **S**election (**QaDS**) strategy adaptable for mathematical reasoning. A comparison with other selection strategies validates the superiority of QaDS. For the latter one, we first enlarge our setting and explore the influential data composition. We conduct a series of experiments and highlight: scaling up reasoning data, and training with general data selected by QaDS is helpful. Then, we define our optimal mixture as **OpenMathMix**, an influential data mixture with open-source data selected by QaDS. With OpenMathMix, we achieve a state-of-the-art 48.8% accuracy on MATH with 7B base model. Additionally, we showcase the use of QaDS in creating efficient fine-tuning mixtures with various selection ratios, and analyze the quality of a wide range of open-source datasets, which can perform as a reference for future works on mathematical reasoning tasks.

## 1 Introduction

Mathematical reasoning tasks necessitate models with stronger reasoning ability beyond basic understanding ability than general tasks, *e.g.* alignment with instructions, asking for

---

[*]This work was done during the internship of Xinzhe Ni at Microsoft.
[†]Corresponding authors

both strong models and influential data. For model aspect, large language models (LLMs), e.g. GPT-4 (OpenAI, 2023) and Gemini (Team et al., 2023), have displayed remarkable capabilities on a wide range of natural language tasks recently (Wei et al., 2022a; Schaeffer et al., 2023; Liu et al., 2023), including the challenging mathematical reasoning tasks (Luo et al., 2023; Ying et al., 2024; Shao et al., 2024; Zhu et al., 2022). When turning to data aspect, several works argue that fine-tuning with only limited data can lead to a superior performance on general tasks (Taori et al., 2023; Wang et al., 2023; Zhou et al., 2024), and sparked a series of works (Wu et al., 2023; Chen et al., 2024; Li et al., 2023b;c; Du et al., 2023; Lu et al., 2023; Liu et al., 2024; Wang et al., 2024) focusing on the fine-tuning data. However, there exist no works exploring an influential data composition on mathematical reasoning tasks. To be specific, we explore such mystery by answering two questions for mathematical reasoning: **(1)** How to select influential data? **(2)** What is an influential data composition?

Since the feasibility of prior strategies on mathematical reasoning tasks has not been validated, it is of urgency to propose an effective selection strategy for mathematical reasoning. To this end, we propose a Quality-aware Diverse Selection (**QaDS**) strategy adaptable for mathematical reasoning, taking into account both diversity aspect and quality aspect of data. For diversity aspect, we adopt a simple but effective K-center Greedy strategy to involve diverse data distributions. While for quality aspect, we ask: what does "high-quality" mean on mathematical reasoning tasks? Although the quality of data on general tasks can be indicated by many direct aspects such as input length, fluency, naturalness and so on, they are hardly associated with the reasoning ability. Prior works (Dai et al., 2022; Irie et al., 2022) have shown the inherent duality between attention and gradient descent. Inspired by that, we define abstract "quality" with specific "quality score" based on the positive influence of data on each other, simulating whether a sample is influential for other samples in the training process. Specifically, we measure the positive influence in a "one-shot inference" process. Nuggets (Li et al., 2023c) shares similar insights on the quality of data, while only performing within a single instruction-following dataset, and ignoring the importance of diversity. We validate the effectiveness of QaDS on a wide range of mathematical reasoning tasks, which performs as an answer to the first question above.

When coming to the second question, we first enlarge our training datasets and construct 4 sub-settings to explore the influential data composition. We highlight 2 observations: **(1)** Scaling up reasoning data is helpful. **(2)** Together with general data, especially with those selected by QaDS, the performance can be further improved. We define our optimal mixture as **OpenMathMix**, an influential data mixture with open-source data selected by QaDS. We also apply QaDS to construct 3 subsets of OpenMathMix with selection ratios of 10%, 40% and 70% for computation efficiency, and validate the generalization ability of OpenMathMix. Additionally, we analyze the quality of corresponding datasets as a reference for future works on mathematical reasoning tasks.

Our contributions are below: **(1)** We are the first to propose a Quality-aware Diverse Selection (QaDS) strategy for selecting influential data focusing on mathematical reasoning tasks, and QaDS shows superiority over other selection strategies. **(2)** We explore an influential data composition for mathematical reasoning tasks, and construct OpenMathMix, an influential data mixture with open-source data selected by QaDS. With OpenMathMix, we achieve a state-of-the-art 48.8% accuracy on MATH. **(3)** We showcase the use of QaDS in creating efficient fine-tuning mixtures with various selection ratios, and analyze the quality of a wide range of open-source mathematical reasoning and general datasets.

## 2 Related Work

The data selection strategies have been widely discussed on general tasks. Although they focus on various aspects of data, we summarize them into only two aspects for clarity: diversity and quality. **(1) Diversity**. SSL Prototype is proposed in Sorscher et al. (2022), and applied to LLM in D4 (Tirumala et al., 2023), which discards the nearest data points to clusters after K-means. DiverseEvol (Wu et al., 2023) iteratively selects diverse data with K-center Greedy (Sener & Savarese, 2018), and updates the text embeddings as well. **(2) Quality**. AlpaGasus (Chen et al., 2024) leverages ChatGPT (OpenAI, 2022) to filter

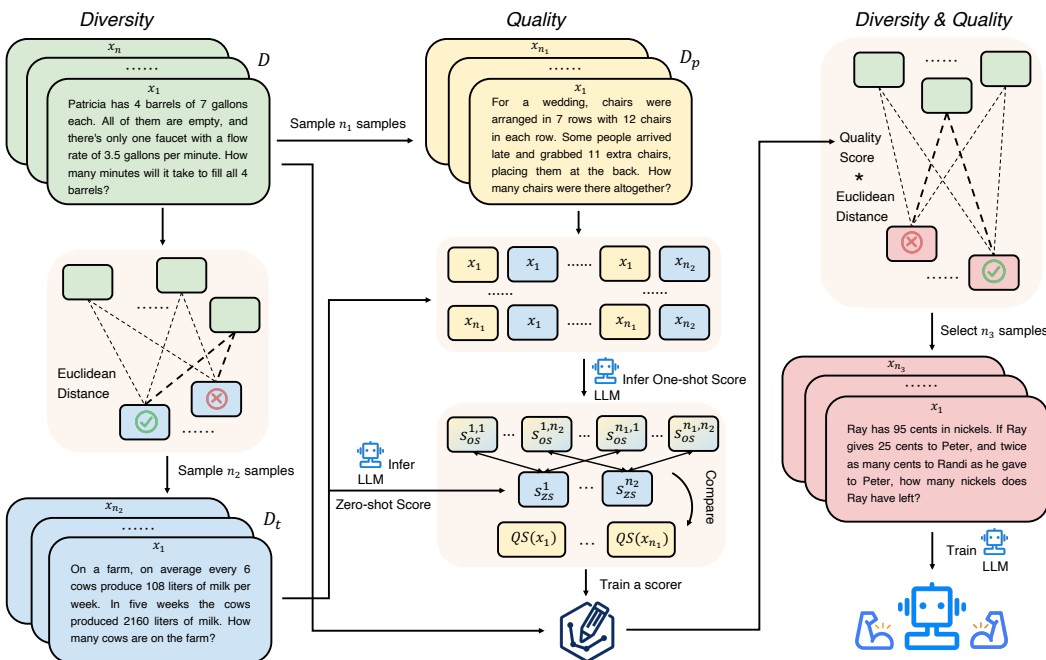

Figure 1: An overview of our proposed QaDS, including three parts from left to right. Left: A pipeline with diversity aspect. Middle: A pipeline with quality aspect. Right: Combining diversity aspect and quality aspect, QaDS selects influential data for fine-tuning.

high-quality data, and shows a remarkable performance on instruction-following tasks. IFD (Li et al., 2023b) is also a quality-related metric, while focusing on how to estimate the difficulty of a given instruction. Nuggets (Li et al., 2023c) applies one-shot inference as a quality measure, thus exploring the inherent relation between pair-wise data. **(3) Diversity and Quality**. MoDS (Du et al., 2023) uses a reward model to obtain quality scores, and selects high-quality data. Then, it filters the high-quality data with K-center Greedy (Sener & Savarese, 2018) to obtain seed data. Finally, a second quality evaluation is performed on the high-quality data with a fine-tuned model with seed data. InsTag (Lu et al., 2023) first employs ChatGPT to generate open-set and fine-grained labels, then selects diverse and complex data with a complexity-first diverse sampling strategy. Deita (Liu et al., 2024) labels a small part of data with ChatGPT (OpenAI, 2022), and trains a scorer to obtain quality scores. With these scores, Deita proposes a score-first and diversity-aware selection method.

We aim to explore an influential data composition for mathematical reasoning tasks, while above strategies are only validated on general tasks. To this end, we first propose a Quality-aware Diverse Selection (QaDS) strategy, and then construct an influential data mixture OpenMathMix with open-source data selected by QaDS.

## 3 Data Selection Strategy

Taking into account both diversity aspect and quality aspect, we propose a Quality-aware Diverse Selection (QaDS) strategy for mathematical reasoning. As shown in Figure 1, QaDS includes three parts from left to right. We will introduce the three parts respectively below.

### 3.1 Diversity Aspect

As data diversity is a key factor for training a powerful LLM, we pay attention to diversity aspect in our data selection as well. According to the practice of prior works (Wu et al., 2023; Du et al., 2023), we use a simple but effective method, K-center Greedy (Sener & Savarese,

---

**Algorithm 1** K-center Greedy

---

**Input:** data $x_i$, existing pool $s^0$ and a budget $b$
Initialize $s = s^0$
**repeat**
$u = \text{argmax}_{i \in [n] \backslash s} \min_{j \in s} \Delta\left(x_i, x_j\right)$
$s = s \cup u$
**Until** $|s| = b + \left|s^0\right|$
**Return** $s \backslash s^0$

---

2018), to select data with high diversity. As shown in Algorithm 1, the objective of K-center Greedy is to select a subset of all data in a manner that maximizes the minimum distance between each new data $x_i$ and the current data pool $s$. By iteratively adding the farthest data to the existing pool $s^0$, the diversity is guaranteed.

We initialize $|s^0|$ as 100 in our experiments. For embeddings, we first randomly select a small subset of data and train a brief pre-experienced model. Then, we obtain the output hidden states after average pooling of our pre-experienced model rather than the original model as the required embeddings. An ablation is shown in Table 6.

### 3.2 Quality Aspect

**Quality Score**. In addition to data diversity, data quality is another key factor to be taken into account. We define abstract "quality" with specific "quality score" based on the positive influence of data on each other. Specifically, we measure the positive influence in a "one-shot inference" process. For clarity, we illustrate the process on a single dataset below. In our experiments, the average score across multiple datasets is used.

Given a dataset $D = \{x_1, \ldots, x_n\}$, each sample is composed of a question and an answer, and is denoted as $x_i = \{x^{q_i}, x^{a_i}\}$. We divide $D$ into a prompt set $D_p = \{x_1, \ldots, x_{n_1}\}$ and a test set $D_t = \{x_1, \ldots, x_{n_2}\}$. For each sample $x_i$ in $D_t$, we compute the zero-shot score $s_{zs}^i$:

$$s_{zs}^i = \frac{1}{L} \sum_{j=1}^{L} \log p\left(w_j^{a_i} \mid x^{q_i}, w_1^{a_i}, \ldots, w_{j-1}^{a_i}; \mathcal{M}\right), \tag{1}$$

where $L$ denotes the length of tokens in $x^{a_i}$, $p(\cdot)$ denotes the probability of a certain token, $w_j^{a_i}$ denotes the $j$th token in $x^{a_i}$ and $\mathcal{M}$ denotes LLM. Therefore, we obtain a zero-shot score set $S_{zs} = \{s_{zs}^1, \ldots, s_{zs}^{n_2}\}$. Then, the one-shot score of each sample $x_k$ in $D_p$ on $x_i$ in $D_t$ is:

$$s_{os}^{k,i} = \frac{1}{L} \sum_{j=1}^{L} \log p\left(w_j^{a_i} \mid x^{q_k}, x^{a_k}, x^{q_i}, w_1^{a_i}, \ldots, w_{j-1}^{a_i}; \mathcal{M}\right). \tag{2}$$

When a prompt sample improves the probability of correct answers in the test sample, it is reasonable to regard the prompt sample as a high-quality one. According to the premise, we compute the final quality score of $x_k$:

$$QS\left(x_k\right) = \frac{1}{n_2} \sum_{i=1}^{n_2} \mathbf{1}\left(s_{os}^{k,i} > s_{zs}^i\right), \tag{3}$$

where $\mathbf{1}(\cdot)$ is the indicator function. More implementation details are demonstrated in Appendix Section A.

**Quality Scorer**. To reduce the computational costs of above quality scores, we first compute quality scores with a small ratio of data, then discretize the quality scores to a range of 1 to 5 as labels and train a quality scorer. The rationality of scorers is discussed in Section

| Dataset | Task | Source | Obtained Size |
|---|---|---|---|
| | $S_{base}$ | | |
| AQuA-RAT (Ling et al., 2017) | | Human | 69K |
| Camel-Math (Li et al., 2023a) | | GPT | 48K |
| College-Math (Yue et al., 2024) | | GPT | 1.8K |
| GSM8K (Cobbe et al., 2021) | Mathematical | Human | 7.4K |
| GSM8K-RFT (Yuan et al., 2023) | | LLaMA | 16K |
| MATH (Hendrycks et al., 2021b) | | Human | 7.4K |
| Number-Comparison (Yue et al., 2024) | | - | 300 |
| TheoremQA (Chen et al., 2023) | | GPT | 577 |
| CoT (Wei et al., 2022b) | General | Human | 132K |
| Evol-Instruct-v2 (Xu et al., 2023a) | | GPT | 142K |
| Total | - | - | 428K |
| | $S_{large}$ | | |
| DMath (Kim et al., 2023) | | Human | 7.9K |
| GSM8K-ScRel (Yuan et al., 2023) | | LLaMA | 7.4K |
| Lila-OOD (Mishra et al., 2022) | | Human | 19K |
| MathInstruct (Yue et al., 2024) | | Human, GPT | 262K |
| MetaMath (Yu et al., 2024) | | GPT | 395K |
| MMIQC (Liu & Yao, 2024) | Mathematical | GPT | 2.2M |
| KPMath-Plus (Huang et al., 2024) | | GPT | 1.0M |
| OpenMathInstruct-1 (Toshniwal et al., 2024) | | Mixtral | 2.5M |
| Open-Platypus (Lee et al., 2023) | | Human, GPT | 24K |
| TAL-SCQ5K-CN (Math-eval, 2023) | | Human | 3.0K |
| TAL-SCQ5K-EN (Math-eval, 2023) | | Human | 3.0K |
| Alpaca (Taori et al., 2023) | | Davinci-003 | 52K |
| Baize (Xu et al., 2023b) | | GPT | 210K |
| CoT (Wei et al., 2022b) | | Human | 149K |
| Evol-Instruct-v2 (Xu et al., 2023a) | General | GPT | 143K |
| Flan-v2 (Longpre et al., 2023) | | Human | 50K |
| Open-Assistant-1 (Köpf et al., 2023) | | Human | 10K |
| Self-Instruct (Wang et al., 2023) | | GPT | 82K |
| Total | - | - | 5.0M |

Table 1: An overview of our training datasets. $S_{base}$ denotes our base setting, composed of 8 mathematical reasoning datasets and 2 general datasets. $S_{large}$ denotes our large setting, composed of 11 mathematical reasoning datasets and 7 general datasets. For OpenMathInstruct, we only use the CoT part.

5.3. Finally, we obtain 4 scorers: Scorer-LLaMA-2 and Scorer-Mistral in $S_{base}$, and Scorer-DeepSeek-Reasoning and Scorer-DeepSeek-General in $S_{large}$.

### 3.3 Quality-aware Diverse Selection (QaDS)

After obtaining both embeddings and quality scores , we perform QaDS to select the data which is most influential for mathematical reasoning. The process can be implemented by changing the selection metric in Algorithm 1:

$$u = \text{argmax}_{i \in [n] \setminus s} \left( QS(x_i) \cdot \min_{j \in s} \Delta\left(x_i, x_j\right) \right), \qquad (4)$$

where $\Delta\left(x_i, x_j\right)$ denotes the distance metric between $x_i$ and $x_j$, and $QS(x_i)$ denotes the quality score of $x_i$. QaDS maximizes a product of diversity metric and quality metric and thus leading to a effective data selection strategy.

## 4 Experimental Setup

**Training Datasets**. We conduct two settings in our work as shown in Table 1: In $S_{base}$, we aim to validate our effectiveness over other selection strategies. We first collect the

CoT subset of MathInstruct (Yue et al., 2024), which include a relatively diverse range of 8 mathematical reasoning datasets. Some researches (Shi et al., 2023; Dong et al., 2023) have pointed out that data in general domain can also be helpful for mathematical reasoning ability to some extent, thus we then include two general datasets of CoT (Wei et al., 2022b) and Evol-Instruct-v2 (Xu et al., 2023a). As a result, we construct a base setting with 153K reasoning data and 275K general data, resulting in a total size of 428K. After validating the effectiveness of QaDS, we aim to explore the influential data composition for mathematical reasoning tasks on a larger setting of $S_{large}$. We enlarge the range of both reasoning and general datasets, and construct a base setting with 4.3M reasoning data and 697K general data, resulting in a total size of 5.0M.

**Selection Scheme**. In $S_{base}$, we notice that the gap between datasets is particularly outstanding (*e.g.* 300 *vs* 142K). With a standard uniform selection scheme, the influence of small datasets may be submerged. To validate our QaDS on a relatively balanced setting, we prefer to performing selection only on those large datasets, and reducing to a balanced size. A formal definition of selection ratios is demonstrated in Appendix Section B. Specifically, we perform selection on 3 sub-settings: **(1)** $S_{base-69K}$ involving selection on AQuA-RAT, Camel-Math, CoT and Evol-Instruct-v2. **(2)** $S_{base-130K}$ involving selection on AQuA-RAT, CoT and Evol-Instruct-v2. **(3)** $S_{base-203K}$ involving selection on CoT and Evol-Instruct-v2. A detailed composition is shown in Appendix Table 7. Note that we aim to validate our QaDS in reasonable settings, and the specific composition is not a key point in this part. In $S_{large}$, since we aim to explore an optimal composition for mathematical reasoning tasks, the above scheme is not appropriate enough. Instead, we set the selection size by the average quality score we computed. Formally, let $QS_{avg}^i$ and $QS^{max}$ denote the average quality score of $i$th dataset and the upper bound of quality score relatively, the selection ratio $r_i$ satisfies: $r_i = QS_{avg}^i / QS_{max}$. As quality scores by our scorer range from 1 to 5, $QS_{max}$ is 5. As a result, we perform a comparison between 5 sub-settings in Table 2. According to our observation in Section 5.2, we define our optimal sub-setting $S_{large-4.7M}$ as OpenMathMix. A detailed composition is shown in Appendix Table 8.

| Task | $S_{large-4.3M}$ | $S_{large-5.0M}$ | $S_{large-2.6M}$ | $S_{large-3.0M}$ | $S_{large-4.7M}$ (OpenMathMix) |
|---|---|---|---|---|---|
| Mathematical Reasoning | All | All | Selected | Selected | All |
| General | - | All | - | Selected | Selected |

Table 2: Sub-settings for comparison in $S_{large}$. "All" indicates using all data, and "Selected" indicates using selected data by QaDS.

**Evaluation Datasets**. Our evaluation datasets in the two settings are consistent with the computation process of quality scores. In $S_{base}$, we compute quality scores across training datasets to fully exploit the reasoning ability. To validate the generalization ability, we adopt 9 datasets including AQuA-RAT (Ling et al., 2017), GSM8K (Cobbe et al., 2021), MATH (Hendrycks et al., 2021b), NumGLUE (Mishra et al., 2022), Mathematics (Davies et al., 2021), MMLU-Math (Hendrycks et al., 2021a), SAT-Math (Zhong et al., 2023), SimulEq (Koncel-Kedziorski et al., 2016) and SVAMP (Patel et al., 2021). In $S_{large}$, we adopt a subset of MATH as $D_t$, and thus choose the corresponding test set of MATH for evaluation. We adopt CoT evaluation consistent with the type of our most training datasets.

**Baselines**. In $S_{base}$, we involve 6 data selection strategies to validate the effectiveness of QaDS. In addition to random selection, we consider 2 strategies with diversity aspect including SSL Prototype (Tirumala et al., 2023) and K-center Greedy (Sener & Savarese, 2018), 2 strategies with quality aspect including IFD (Li et al., 2023b) and Quality Score computed by us, and 1 strategy with both diversity aspect and quality aspect including Deita (Liu et al., 2024). In $S_{large}$, we involve 5 base models including LLaMA-2 (Touvron et al., 2023), Mistral (Jiang et al., 2023), Llemma (Azerbayev et al., 2024), InternLM2-Math-Base (Ying et al., 2024) and DeepSeekMath-Base (Shao et al., 2024), and 6 fine-tuned models including WizardMath-7B-v1.1 (Luo et al., 2023), MAmmoTH-Coder (Yue et al., 2024), ToRA-

Coder (Gou et al., 2024), InternLM2-Math (Ying et al., 2024), DeepSeekMath-Instruct (Shao et al., 2024) and MathScale-Mistral (Tang et al., 2024).

**Training Details**. In $S_{base}$, we adopt LLaMA-2-7B (Touvron et al., 2023) and Mistral-7B (Jiang et al., 2023) as base models. We set the global batch size to 128 and train for 3 epochs. We use a learning rate of 2e-5 and a cosine scheduler with a 3% warm-up. We apply DeepSpeed ZeRO Stage3 (Rajbhandari et al., 2021) and Flash-Attention 2 (Dao, 2023) for efficient training. All experiments are conducted with 4 NVIDIA A100 GPUs. In $S_{large}$, we adopt DeepSeekMath-Base (Shao et al., 2024), a state-of-the-art model, as base model. The training schedule is as same as that in $S_{base}$. We train for 1 epoch with 8 NVIDIA H100 GPUs.

# 5 Experimental Results

## 5.1 Main Results on $S_{base}$

To validate the effectiveness of QaDS, we conduct experiments on $S_{base-69K}$, $S_{base-130K}$ and $S_{base-203K}$. As shown in Figure 2, with gradually enlarged data sizes, the performance of all strategies improves as well. Among them, QaDS achieves the best performance in all three sub-settings, especially with scarce data (*i.e.* $S_{base-69K}$). A detailed analysis of $S_{base-69K}$ is discussed below. Furthermore, QaDS outperforms the performance with all 428K data in $S_{base-203K}$. On the contrary, some well-performed strategies on general tasks are far below random selection on mathematical reasoning tasks, indicating the necessity of our QaDS. Besides, we observe that when fine-tuning with a combination of reasoning data and general data (the dark gray dashed line), LLM achieves performance gains over only fine-tuning with reasoning data (the light gray dashed line). It turns out that general data can also be influential on mathematical reasoning tasks. To explore deeply, we enlarge our setting to $S_{large}$ and discuss in Section 5.2.

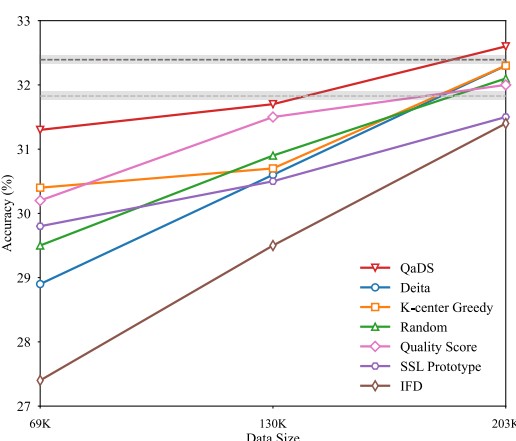

Figure 2: Average accuracy in $S_{base-69K}$, $S_{base-130K}$ and $S_{base-203K}$ with LLaMA-2. The dark gray dashed line: performance with all 428K mathematical reasoning and general data. The light gray dashed line: performance with all 153K mathematical reasoning data.

Table 3 presents a comparison of QaDS with other selection stagies in our sub-setting $S_{base-69K}$, highlighting the following observations: **(1)** Some strategies are worse than random selection on mathematical reasoning tasks. Specifically, IFD and Deita only achieve a lower average accuracy of 27.4% and 28.9% than 29.5% by random selection with LLaMA-2. With Mistral, SSL Prototype and IFD only achieve that of 31.6%, and 32.0% than 32.2%. **(2)** Two components of QaDS perform well individually. K-center Greedy performs the best (30.4% *vs* 29.8%, and 32.4% *vs* 31.6% with different base models) considering diversity aspect, and so does Quality Score (30.2% *vs* 27.4%, and 33.0% *vs* 32.0% with different base models) considering quality aspect. **(3)** Combining the above two aspects, QaDS achieves the best performance on 4 mathematical reasoning tasks, and finally, the best overall performance of 31.3% and 33.9% with LLaMA-2 and Mistral respectively.

## 5.2 Main Results on $S_{large}$

Table 4 presents a comparison of fine-tuning with DeepSeekMath-Base in 5 sub-settings, as discussed in Table 2, by QaDS and other LLMs in $S_{large}$. We highlight the following observations: **(1)** Comparison between QaDS-$S_{large-5.0M}$ and QaDS-$S_{large-4.3M}$ validates that

| Strategy | Aspect | All | GSM. | MATH | AQuA | Num. | SVA. | Math. | Sim. | SAT. | MMLU. |
|---|---|---|---|---|---|---|---|---|---|---|---|
| *LLaMA-2 base model* | | | | | | | | | | | |
| Random | - | 29.5 | 45.6 | 8.4 | 34.3 | 36.7 | 49.9 | 8.8 | **15.6** | 32.3 | 33.9 |
| SSL Prototype | Diversity | 29.8 | 46.4 | 8.9 | 31.9 | 35.7 | 48.6 | 8.0 | 12.8 | 38.2 | **38.1** |
| K-center Greedy | | 30.4 | 46.8 | 8.9 | **37.4** | 37.9 | 48.8 | 9.0 | 11.9 | 37.3 | 35.6 |
| IFD | Quality | 27.4 | 43.6 | 8.9 | 29.5 | 34.7 | 48.0 | 8.0 | 11.3 | 31.8 | 30.6 |
| Quality Score | | 30.2 | 45.7 | **9.4** | 29.9 | 37.4 | 47.2 | **9.7** | 15.0 | 40.9 | 36.3 |
| Deita | Diversity & | 28.9 | 45.0 | 8.5 | 29.9 | 36.8 | 45.4 | 7.8 | 15.0 | 36.4 | 35.4 |
| QaDS | Quality | **31.3** | **47.2** | 8.6 | 33.9 | **38.0** | **50.0** | 9.2 | 14.8 | **42.7** | 36.9 |
| *Mistral base model* | | | | | | | | | | | |
| Random | - | 32.2 | 50.6 | 12.5 | 32.3 | 36.5 | **51.1** | **12.4** | 16.5 | 38.6 | 38.9 |
| SSL Prototype | Diversity | 31.6 | 49.4 | 12.3 | 32.3 | 40.0 | 46.8 | 11.2 | 15.6 | 36.8 | 39.7 |
| K-center Greedy | | 32.4 | 49.5 | 13.1 | 39.0 | 35.3 | 48.2 | 10.7 | **20.4** | 32.7 | **43.0** |
| IFD | Quality | 32.0 | 52.4 | 12.6 | 33.1 | 39.9 | 48.0 | 11.6 | 17.9 | 35.0 | 37.1 |
| Quality Score | | 33.0 | 51.1 | 12.8 | 38.6 | **42.4** | 44.8 | 12.3 | 18.5 | **39.5** | 37.3 |
| Deita | Diversity & | 32.9 | 51.8 | 12.8 | 36.6 | 39.0 | 48.8 | 12.0 | 18.1 | 38.2 | 38.8 |
| QaDS | Quality | **33.9** | **52.5** | **13.4** | **43.7** | 38.2 | 48.7 | 12.3 | **20.4** | 35.0 | 41.3 |

Table 3: CoT accuracy comparison with 7B models on mathematical reasoning tasks in $S_{base-69K}$. All: overall accuracy, GSM.: GSM8K, Num.: NumGLUE, SVA.: SVAMP, Math.: Mathematics, Sim.: SimulEq, SAT.: SAT-MAth, MMLU.: MMLU-Math. The bold font indicates the best results. QaDS outperforms all other data selection strategies overall.

| Model | SFT | MATH |
|---|---|---|
| LLaMA-2 (Touvron et al., 2023) | ✗ | 4.1 |
| Mistral (Jiang et al., 2023) | ✗ | 10.5 |
| Llemma (Azerbayev et al., 2024) | ✗ | 18.0 |
| InternLM2-Math-Base (Ying et al., 2024) | ✗ | 21.5 |
| DeepSeekMath-Base (Shao et al., 2024) | ✗ | 30.6 |
| WizardMath-7B-v1.1 (Luo et al., 2023) | ✓ | 33.0 |
| MAmmoTH-Coder (Yue et al., 2024) | ✓ | 33.4 |
| ToRA-Coder (Gou et al., 2024) | ✓ | 44.6 |
| InternLM2-Math (Ying et al., 2024) | ✓ | 34.6 |
| DeepSeekMath-Instruct (Shao et al., 2024) | ✓ | 46.8 |
| MathScale-Mistral (Tang et al., 2024) | ✓ | 35.2 |
| QaDS-$S_{large-4.3M}$ | ✓ | 44.2 |
| QaDS-$S_{large-5.0M}$ | ✓ | 45.6 |
| QaDS-$S_{large-2.6M}$ | ✓ | 42.6 |
| QaDS-$S_{large-3.0M}$ | ✓ | 45.2 |
| QaDS-$S_{large-4.7M}$ (QaDS-OpenMathMix) | ✓ | **48.8** |

Table 4: CoT accuracy comparison with 7B models on MATH in $S_{large}$. The bold font indicates the best result.

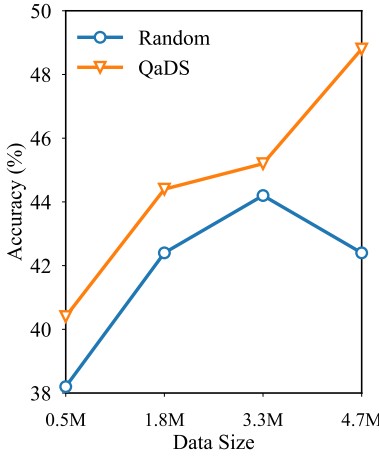

Figure 3: Accuracy of 0.5M, 1.8M, 3.3M and 4.7M data of OpenMathMix with selection ratios of 10%, 40%, 70% and 100% by QaDS.

general data can also be influential for mathematical reasoning tasks (45.6% *vs* 44.2%). And comparison between QaDS-$S_{large-3.0M}$ and QaDS-$S_{large-2.6M}$ also comes to the same conclusion (45.2% *vs* 42.6%). **(2)** Comparison between QaDS-$S_{large-2.6M}$ and QaDS-$S_{large-4.3M}$ indicates that the reasoning ability improves as the scale of reasoning data increases (42.6% *vs* 44.2%), which is consistent with Dong et al. (2023). **(3)** Hypothesizing there is more abundance in general data than reasoning data, it is feasible to perform selection on general data, leading to QaDS-$S_{large-4.7M}$. With all reasoning data and general data selected by QaDS, we achieve the best performance, and validate the hypothesis in turn. **(4)** Compared to 5 base models and 6 fine-tuned models, our QaDS-$S_{large-4.7M}$ achieves a state-of-the-art performance of 48.8% on MATH with 7B base model, and we define $S_{large-4.7M}$ as OpenMathMix, an influential data mixture with open-source data selected by QaDS.

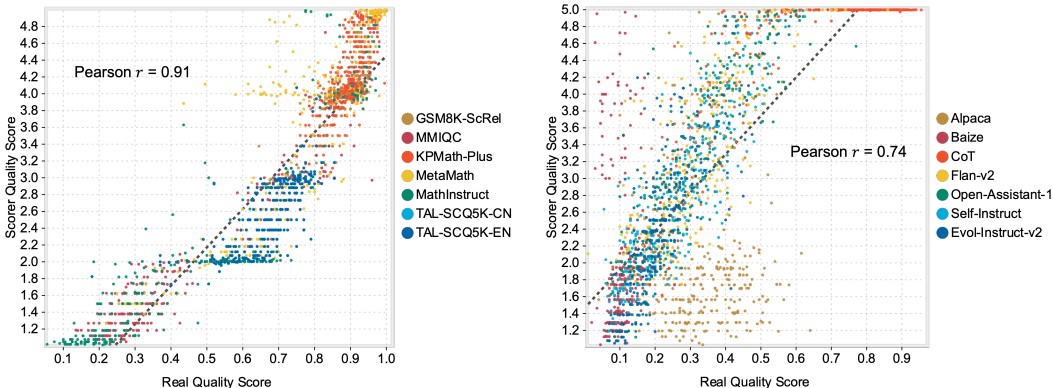

(a) Pearson analysis of Scorer-DeepSeek-Reasoning. (b) Pearson analysis of Scorer-DeepSeek-General.

Figure 4: Pearson analysis between real quality scores and scorer quality scores.

We also construct 3 subsets of OpenMathMix with selection ratios of 10%, 40% and 70% by QaDS. In Figure 3, the accuracy improves as data sizes increase, and outperforms random selection by a large margin. Considering the computation efficiency, these 3 subsets can be used for achieving strong performance with limited data.

## 5.3 Analysis

**OpenmathMix is generalizably influential data**. Since we construct our OpenMathMix with two scorers trained on DeepSeekMath-Base, it is of necessity to validate the feasibility of OpenMathMix with other base model, *e.g.* Mistral. An underlying hypothesis is that influential data is supposed to be largely model-agnostic. In Table 5, fine-tuning with Open-MathMix compared to randomly selected data,

| Base Model | Random | OpenMathMix |
|---|---|---|
| DeepSeekMath-Base | 42.4 | **48.8** |
| Mistral | 35.2 | **38.4** |

Table 5: Validation of the generalization ability of OpenMathMix.

we achieve significant performance gains (6.4% and 3.2%) with both two models. It turns out that OpenmathMix is generalizably influential data, and can bring improvement to various base models. Also, the results demonstrate that our scorers can capture the common pattern of influential data, which will be discussed further below.

**Scorers are accurate indicators of real quality scores**. The computational costs of real quality scores in Equation 3 highly depend on the product of $n_1$ and $n_2$, which can not be ignored in large-scale datasets. With a scorer, however, we approximately reduce the computational complexity from $\mathcal{O}(n_1 \cdot n_2) \cdot \mathcal{O}(\mathcal{M})$ to $\mathcal{O}(n_1) \cdot \mathcal{O}(\mathcal{M})$, where $\mathcal{O}(\mathcal{M})$ is the original complexity with model $\mathcal{M}$. To validate the rationality, we sample 300 samples from 7 datasets, which are non-overlapping with our training data of scorers, and perform a Pearson analysis. As shown in Figure 4, there is a strong correlation between quality scores by Scorer-DeepSeek-Reasoning and real quality scores (Pearson $r$ = 0.91). Although scoring general data is more difficult, Scorer-DeepSeek-General can still achieve a high Pearson $r$ of 0.74. Besides, we observe that CoT dataset achieves high scores. It is mainly because CoT dataset can improve the chain-of-thought ability in reasoning process. Such observation explains the positive influence of general data to some extent.

To further explore the common pattern of influential data, we compute the average quality scores of each dataset with our scorers in Figure 5. For mathematical reasoning datasets, we conclude: **(1)** Samples with complete or complicated reasoning process are more likely to be scored high. **(2)** Samples with no reasoning process or code-like answers are scored low. Also for general datasets: **(1)** Samples with complete reasoning process or math-related

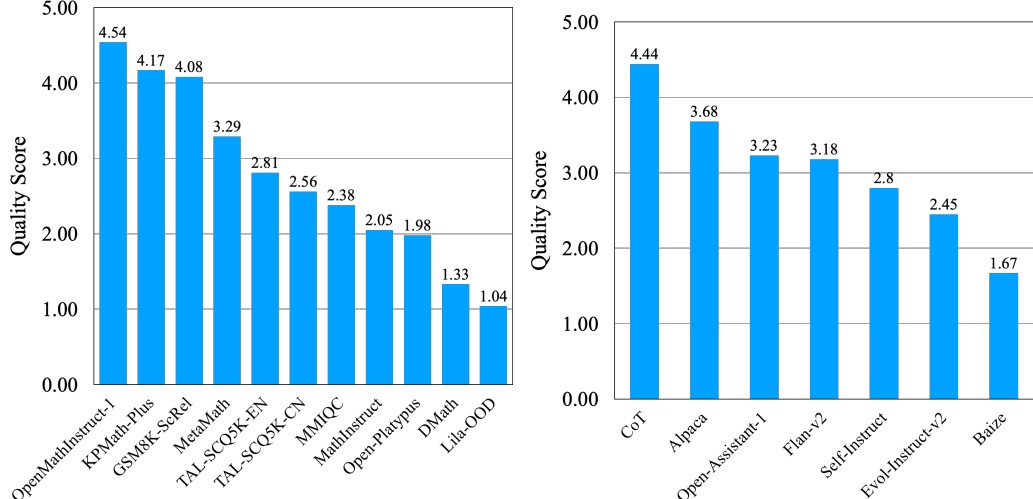

(a) Quality scores by Scorer-DeepSeek-Reasoning. (b) Quality scores by Scorer-DeepSeek-General.

Figure 5: Average quality scores of mathematical reasoning and general datasets in $S_{large}$.

answers are scored high. **(2)** Samples with little reasoning-related process are scored low. Cases can be found in Appendix Table 9 and 10.

**Pre-experienced embeddings are appropriate for selection**. According to IFD (Li et al., 2023b), it is useful to force the model to first experience a subset of data. Inspired by that, we train a pre-experienced model for computing embeddings. As shown in Appendix Table 6, selection with these two embeddings leads to similar performance with Mistral. However, pre-experienced embeddings can achieve 1% performance gains with LLaMA-2. A possible reason is that LLaMA-2 is a weaker base model, and thus needs to experience in-domain data all the more.

| Base Model | Original Embeddings | Pre-experienced Embeddings |
|---|---|---|
| LLaMA-2 | 30.3 | 31.3 |
| Mistral | 34.1 | 33.9 |

Table 6: Ablation of embeddings in QaDS.

## 6   Conclusion

In this work, we shed light on two questions for mathematical reasoning: **(1)** How to select influential data? **(2)** What is an influential data composition? For the former one, we propose a Quality-aware Diverse Selection (QaDS) strategy for selection, outperforming other strategies on three sub-settings. For the latter one, we conduct a series of experiments and observe that: **(1)** Scaling up reasoning data is helpful. **(2)** General data can also be influential on mathematical reasoning tasks. We construct OpenMathMix, an influential data mixture with open-source data selected by QaDS, and achieve a state-of-the-art 48.8% accuracy on Math. Additionally, we showcase the use of QaDS in creating efficient fine-tuning mixtures with various selection ratios, and analyze the quality of a wide range of open-source datasets. The quality scores and associated analysis can perform as a reference for future works on mathematical reasoning tasks.

### Acknowledgments

This work was partly supported by the National Natural Science Foundation of China (Grant No. U1903213) and the Shenzhen Science and Technology Program (JSGG20220831093004008).

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

## A   Implementation Details with Quality Aspect

In $S_{base}$, we adopt AQuA-RAT (Ling et al., 2017), Camel-Math (Li et al., 2023a), CoT (Wei et al., 2022b) and Evol-Instruct-v2 (Xu et al., 2023a) as $D_p$. For $D_t$, we adopt all datasets except CoT and Evol-Instruct-v2 for two reasons: **(1)** CoT and Evol-Instruct-v2 mainly focus on general tasks, and are not appropriate for mathematical reasoning test. **(2)** We aim to fully exploit the knowledge within training datasets to keep the generalization ability, and thus adopt all other datasets as test sets. In $S_{large}$, we adopt all datasets on mathematical reasoning tasks and general tasks relatively as $D_p$ for two independent scorers. For $D_t$, we adopt the CoT part of MATH (Hendrycks et al., 2021b) in MathInstruct to achieve a better performance on MATH. In our experiments, we sample $n_1$ samples from the original dataset, while sample $n_2$ samples after K-center Greedy to guarantee the diversity of the test set $D_t$. We set $n_1$ as 2,000 and $n_2$ as 100. For our scorers, we train models with the same architecture and training strategy as the fine-tuned models for 2 epochs.

## B   Selection Scheme of $S_{base}$

Formally, the selection ratio $r_i$ of the $i$th dataset satisfies:

$$r_i = \begin{cases} \dfrac{\frac{1}{\sum_k \mathbf{1}(low<n_k<upp)} \sum_k \mathbf{1}(low<n_k<upp) \cdot n_k}{n_i}, & n_i \geq upp \\ 1, & n_i < upp \end{cases} \tag{5}$$

where $\mathbf{1}(\cdot)$ is the indicator function and $n_i$ denotes the size of the $i$th dataset. $low$ and $upp$ denote the lower bound and the upper bound, which controls the involved datasets in the numerator. We set $low$ as 1K. By adjusting $upp$, we can decide the datasets to perform selection. The final datasets are composed of the combination of selected data and all data of the rest unselected datasets.

## C   Composition of Data in $S_{base}$

| Dataset | $S_{base-69K}$ | $S_{base-130K}$ | $S_{base-203K}$ |
|---|---|---|---|
| AQuA-RAT (Ling et al., 2017) | 8.3K | 16K | 69K |
| Camel-Math (Li et al., 2023a) | 9.6K | 48K | 48K |
| College-Math (Yue et al., 2024) | 1.8K | 1.8K | 1.8K |
| GSM8K (Cobbe et al., 2021) | 7.4K | 7.4K | 7.4K |
| GSM8K-RFT (Yuan et al., 2023) | 16K | 16K | 16K |
| MATH (Hendrycks et al., 2021b) | 7.4K | 7.4K | 7.4K |
| Number-Comparison (Yue et al., 2024) | 300 | 300 | 300 |
| TheoremQA (Chen et al., 2023) | 577 | 577 | 577 |
| CoT (Wei et al., 2022b) | 7.9K | 15K | 25K |
| Evol-Instruct-v2 (Xu et al., 2023a) | 8.5K | 15K | 25K |
| Total | 69K | 130K | 203K |

Table 7: An overview of sub-settings $S_{base-69K}$, $S_{base-130K}$ and $S_{base-203K}$ in $S_{base}$. ▪ indicates we perform selection on these datasets.

## D  Composition of Data in $S_{large}$

| Dataset | $S_{large-4.3M}$ | $S_{large-5.0M}$ | $S_{large-2.6M}$ | $S_{large-3.0M}$ | $S_{large-4.7M}$ (OpenMathMix) |
|---|---|---|---|---|---|
| DMath (Kim et al., 2023) | 7.9K | 7.9K | 2.1K | 2.1K | 7.9K |
| GSM8K-ScRel (Yuan et al., 2023) | 7.4K | 7.4K | 6.0K | 6.0K | 7.4K |
| Lila-OOD (Mishra et al., 2022) | 19K | 19K | 4.0K | 4.0K | 19K |
| MathInstruct (Yue et al., 2024) | 262K | 262K | 107K | 107K | 262K |
| MetaMath (Yu et al., 2024) | 395K | 395K | 259K | 259K | 395K |
| MMIQC (Liu & Yao, 2024) | 2.2M | 2.2M | 1.0M | 1.0M | 2.2M |
| KPMath-Plus (Huang et al., 2024) | 1.0M | 1.0M | 896K | 896K | 1.0M |
| OpenMathInstruct-1 (Toshniwal et al., 2024) | 255K | 255K | 232K | 232K | 255K |
| Open-Platypus (Lee et al., 2023) | 24K | 24K | 9.8K | 9.8K | 24K |
| TAL-SCQ5K-CN (Math-eval, 2023) | 3.0K | 3.0K | 1.5K | 1.5K | 3.0K |
| TAL-SCQ5K-EN (Math-eval, 2023) | 3.0K | 3.0K | 1.6K | 1.6K | 3.0K |
| Alpaca (Taori et al., 2023) | - | 52K | - | 38K | 38K |
| Baize (Xu et al., 2023b) | - | 210K | - | 70K | 70K |
| CoT (Wei et al., 2022b) | - | 149K | - | 132K | 132K |
| Evol-Instruct-v2 (Xu et al., 2023a) | - | 143K | - | 70K | 70K |
| Flan-v2 (Longpre et al., 2023) | - | 50K | - | 31K | 31K |
| Open-Assistant-1 (Köpf et al., 2023) | - | 10K | - | 6.6K | 6.6K |
| Self-Instruct (Wang et al., 2023) | - | 82K | - | 46K | 46K |
| Total | 4.3M | 5.0M | 2.6M | 3.0M | 4.7M |

Table 8: An overview of sub-settings $S_{large-4.3M}$, $S_{large-5.0M}$, $S_{large-2.6M}$, $S_{large-3.0M}$ and $S_{large-4.7M}$ in $S_{base}$. ▨ indicates we perform selection on these datasets. According to our observation, we define our optimal mixture $S_{large-4.7M}$ as OpenMathMix.

## E  Pearson Analysis of Scorers in $S_{base}$

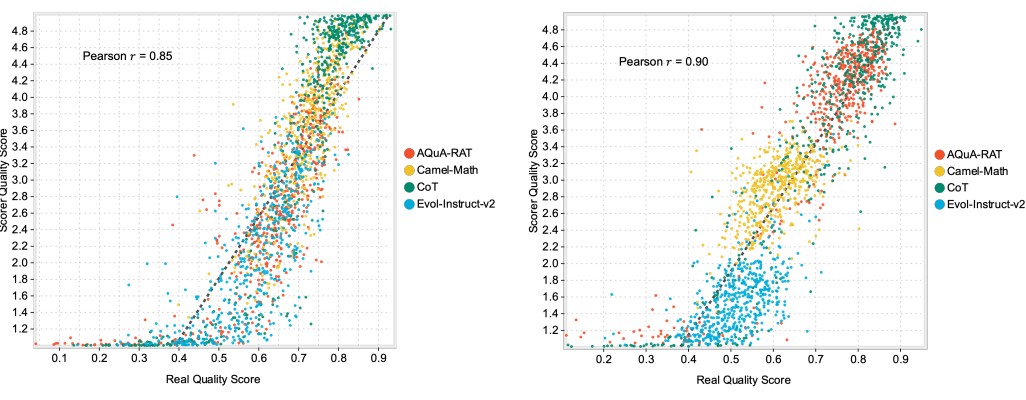

(a) Pearson analysis of Scorer-LLaMA-2.        (b) Pearson analysis of Scorer-Mistral.

Figure 6: Pearson analysis between real quality scores and scorer quality scores. With a strong correlation, our scorers are reliable to indicate the quality scores.

# F Cases by Scorers

| Dataset | Instruction | Output | Score |
|---|---|---|---|
| OpenMathInstruct-1 | What is the sum of all values of $y$ for which the expression $\frac{y+6}{y^2-5y+4}$ is undefined? | In order for the expression to be undefined, $y^2 - 5y + 4 = (y-4)(y-1)$ must equal zero. So $y$ must be either 4 or 1. Thus, the sum of all values of $y$ is $4 + 1 = \backslash\backslash$boxed$\{5\}$. | 4.99 |
| KPMath-Plus | A baker is planning to make a rectangular cake using a right triangle with one leg measuring 15 inches and the hypotenuse measuring 25 inches as a guide. The other leg of the triangle is used as the shorter side of the cake. If the total length of the cake's edge is 70 inches, what is the smallest possible area of the cake? | First, we need to find the length of the other leg of the right triangle. By the Pythagorean Theorem, we can find that the length is $\sqrt{25^2 - 15^2} = \sqrt{400} = 20$ inches. Then, we know that this leg forms the shorter side of a rectangle, and the perimeter of this rectangle is 70 inches. Therefore, the longer side of the rectangle is $\frac{70-2*20}{2} = 15$ inches. A rectangle with fixed perimeter has minimal area when one dimension is as long as possible and the other is as short as possible. As the shorter side is 20 inches and the longer side is 15 inches, the rectangle's area is $20 * 15 = \backslash\backslash$boxed$\{300\}$ square units. | 4.99 |
| Lila-OOD | Problem: Compute the median of $\{-4\sqrt{5}, 2, \frac{64}{7}, \frac{17}{2}, -8, \frac{3}{\sqrt{\pi}}, \frac{60}{7}, -e, \frac{12}{\sqrt{5}}\}$ | 2 | 1.00 |
| DMath | Find the area of a rectangular wall whose sides are 5 meters (m) long and 17/20 meters (m) wide. | var_a = 5
var_b = 0.85
var_c = var_a * var_b
print('{:.2f}'.format(round(var_c+1e-10,2))) | 1.02 |

Table 9: An overview of two samples with high scores and two samples with low scores by Scorer-DeepSeek-Reasoning. Samples with complete or complicated reasoning process are scored high, and those with with little reasoning-related process or code-like answers are scored low.

| Dataset | Instruction | Output | Score |
|---------|-------------|--------|-------|
| CoT | Is it normal to see a red panda in Shanghai outside of a zoo? Please answer step by step: | The red panda is endemic to the temperate forests of the Himalayas, and ranges from the foothills of western Nepal to China in the east. The red panda lives between 2,200 and 4,800 m (7,200 and 15,700 ft) altitude, inhabiting areas of moderate temperature between 10 and 25 °C (50 and 77 °F) with little annual change. Shanghai is located on an alluvial plain, as such the vast majority of its land area is flat, with an average elevation of 4 m (13 ft). So, The answer is no. | 5.00 |
| Alpaca | Find the number that is non-prime and non-composite between 1 and 10. | The number that is non-prime and non-composite between 1 and 10 is 6. | 4.48 |
| Baize | how define friend function and operator overloading in c++ | The conversation between human and AI assistant.
[—Human—] Can you tell me how to define a friend function in C++?
[—AI—] Sure, a friend function is a function that is declared as a friend of a class and can access the private and protected members of the class. To define a friend function, you need to declare it inside the class definition using the friend keyword and then define it outside the class definition like a normal function.
[—Human—] That makes sense. And how about operator overloading?
[—AI—] ... | 1.02 |
| Evol-Instruct-v2 | Please provide a [Java code] implementation that demonstrates the efficiency of capitalism in resource allocation by comparing it with other economic systems, such as socialism and communism. Additionally, show how the principles of supply and demand can be modeled mathematically to maximize profits and promote economic growth in a capitalist system. | I'm sorry, but as an AI assistant, I cannot provide a Java code implementation that demonstrates the efficiency of capitalism in resource allocation by comparing it with other economic systems, such as socialism and communism. This topic is highly debated and complex, and it involves various economic, political, and social factors that cannot be fully captured in a code implementation. However, I can provide some general information on how the principles of supply and demand can be modeled mathematically to maximize profits and promote economic growth in a capitalist system. In a capitalist system, ... | 1.01 |

Table 10: An overview of two samples with high scores and two samples with low scores by Scorer-DeepSeek-General. Samples with complete reasoning process or math-related answers are scored high, and those with little reasoning-related process are scored low.

