# OpenReview forum: "Exploring the Mystery of Influential Data for Mathematical Reasoning"
_colmweb.org/COLM/2024/Conference — COLM_

### Official Review · Reviewer_gRx8 · 2024-05-09

**Rating:** 4
**Confidence:** 4
**Ethics Flag:** 1

**Summary:**

This paper mainly studies data selection for supervised instruction tuning of LLMs, targeted at math reasoning. Specifically, it follows existing data selection work that selects supervised tuning data based on the combination of its diversity and quality. Experiments on large-scale open-source data show that the proposed data selection approaches outperform (not by a big margin) both random selection and other baseline data selection approaches.

**Reasons To Accept:**

1. The experiment on math reasoning is solid, as it evaluates the proposed data selection from multiple perspectives.
2. The paper is relatively easy to understand.

**Reasons To Reject:**

1. My primary concern is the motivation behind the paper, specifically the research question addressing the math reasoning task. While the paper focuses on math reasoning, the proposed method appears universally applicable to various tasks. After a thorough review, I found no clear motivation or adaptation of the method specifically for the challenges in math reasoning. An interesting idea might involve a compositional aspect, such as whether different combinations of formulas could lead to improvements.

2. The reported improvements in results are quite marginal. It is unclear what a ~1.5-2% improvement in math reasoning signifies in this context. Such modest gains might be overshadowed by the capabilities of a base model, such as LLAMA 70B. Consequently, the effectiveness of the proposed approach remains ambiguous.

---

> ### Author Rebuttal · Authors · 2024-05-30
>
> Dear Reviewer gRx8,
>
> Thank you for your thoughtful feedback! We are pleased to share our ideas to your concerns.
>
> > No clear motivation for math.
>
> We emphasize our motivation below:
>
> * Math reasoning task is a **more challenging** task compared to general tasks. We found that **existing strategies do not perform well on reasoning tasks** in paper "Figure 2", and thus focus on math reasoning.
> * If there exists a right and comprehensive **reasoning process** in a certain sample, it should be positive to another sample. We'd like to use the positive influence of reasoning data, leading to the "one-shot inference". The theoretical support is the inherent duality between attention and gradient descent [1].
>
> Also, we conduct experiments on two **general tasks** taking into account your question. Please see the table in our rebuttal for **Reviewer YKSQ**.
>
> > Whether different combinations could lead to improvements.
>
> We perform ablation of the two aspects of QaDS in paper "Table 3", and we order it more clearly below:
>
> |             Strategy             | LLaMA-2 Overall | Mistral Overall |
> | :------------------------------: | :-------------: | :-------------: |
> |              Random              |      29.5       |      32.2       |
> | Only Diversity (K-center Greedy) |      30.4       |      32.4       |
> |   Only Quality (Quality Score)   |      30.2       |      33.0       |
> | Diversity and Quality (**QaDS**) |    **31.3**     |    **33.9**     |
>
> > Gains might be overshadowed by LLAMA 70B.
>
> Firstly, we conduct experiments on LLaMA-2 70B considering your insightful comment:
>
> | Strategy | Overall  | GSM.     |   MATH   | AQuA     |   Num.   | SVA.     |  Math.   |   Sim.   |   SAT.   |  MMLU.   |
> | :------: | :------: | -------- | :------: | -------- | :------: | -------- | :------: | :------: | :------: | :------: |
> |   IFD    |   43.3   | 69.4     |   18.0   | 42.7     |   47.9   | 75.8     |   19.7   |   23.5   |   44.1   |   48.6   |
> |  Deita   |   42.7   | 68.2     |   18.0   | 41.3     | **52.1** | 75.7     | **21.1** |   22.4   | **45.1** |   40.5   |
> | **QaDS** | **44.7** | **71.0** | **18.2** | **46.9** |   48.0   | **77.9** |   20.1   | **29.0** |   42.3   | **48.9** |
>
> Then, we also emphasize that we reach an improvement of 6.6% in paper "Figure 3" (48.8 vs 42.2) in *S_{large}*-4.7M.
>
> [1] Why can gpt learn in-context? language models implicitly perform gradient descent as meta-optimizers. *arXiv preprint arXiv:2212.10559* (2022).

---

> > ### Author Response · Authors · 2024-06-06
> > **Call for Discussion**
> >
> > Dear reviewer gRx8,
> >
> > Thank you very much for taking the time to review our paper and provide valuable feedback. As the discussion period is coming to an end, we would like to ensure that our recent response satisfactorily addresses your concerns. If you have any further concerns, you are welcome to point them out, and we are happy to clarify them further. Thank you!
> >
> > Best，
> > Submission310 Authors

---

### Official Review · Reviewer_YKSQ · 2024-05-12

**Rating:** 7
**Confidence:** 4
**Ethics Flag:** 1

**Summary:**

This paper focuses on selecting influential training data for improving LLMs' mathematical reasoning ability. The authors propose a quality-aware diverse selection strategy (QaDS) over other selection strategies and also define an OpenMathMix which is a data mixture selected by QaDS. The experiments show advantages of the proposed strategy and the paper is comprehensive.

**Questions To Authors:**

Could this strategy be applied to other tasks than the mathematical reasoning?

**Reasons To Accept:**

* Proposed a reasonable way of selecting high-quality data (one-shot better than zero-shot)
* Define an influential data mixture with open-source data selected by QaDS

**Reasons To Reject:**

* Lack of further analysis of the selection strategy, e.g., more than one-shot

---

> ### Author Rebuttal · Authors · 2024-05-28
>
> Dear Reviewer YKSQ,
>
> Thanks for your valuable comments of our paper! We are pleased to share our ideas on your questions.
>
> > Lack of further analysis of the selection strategy, e.g., more than one-shot.
>
> We provide analysis from two aspects to your concern:
>
> * Existing strategies mainly consider the data quality from some direct aspects such as input length, fluency, naturalness and so on, while **not fit for reasoning**. Therefore, we found that they do not perform well on reasoning tasks as shown in paper "Figure 2".
> * The quality of resoning data is not easy to measure, and our solution is to measure **the positive influence on each other**: if there exists a right and comprehensive reasoning process in a certain sample, it should be positive to another sample, leading to the "one-shot inference". The theoretical support is the inherent duality between attention and gradient descent, and can be found in [1].
>
> We are the first to explore data selection for reasoning, and we hope we can provide insights to the community. We will explore more strategies in our future work.
>
> > Could this strategy be applied to other tasks than the mathematical reasoning?
>
> Yes! Although we focus on the challenging mathematical reasoning task, we also conduct experiments on two general tasks taking into account your question.
>
> We first adopt 2000 samples of Alpaca and Dolly datasets as *D_{p}*, and 500 mixing samples of Alpaca and Dolly as *D_{t}* to perform one-shot inference. For the gap between reasoning and general tasks, we then train a new scorer for general tasks with above data. The scoring and selection process is the same as we mentioned in our paper. Finally, we fine-tune LLaMA-2 with our selected data (6.6K from 66K) by QaDS. Also, we implement IFD and Deita to select the same size 6.6K of data. We show results below:
>
> | Strategy | MT-Bench | AlpacaEval (%) |
> | :------: | :------: | :------------: |
> |   IFD    |   5.39   |      56.6      |
> |  Deita   |   5.23   |      49.8      |
> | **QaDS** | **5.49** |    **72.6**    |
>
> It turns out that our QaDS also achieves the best results, showing the effectiveness of our strategy.
>
> [1] Why can gpt learn in-context? language models implicitly perform gradient descent as meta-optimizers. *arXiv preprint arXiv:2212.10559* (2022).

---

### Official Review · Reviewer_oJ7q · 2024-05-13

**Rating:** 7
**Confidence:** 4
**Ethics Flag:** 1

**Summary:**

__Summary__

1. Propose a quality-aware diverse selection approach for mathematical reasoning. This approach aims to integrate both quality and diversity in selecting influential data.
2. Diversity: sampling by maximizing the minimum distance between the data points, which are encoding by embeddings.
3. Quality: the average times that one-shot performance is better than the zero-shot performamce.
     1. They further train a quality scorer for that to reduce the intractable computational cost.
4. Then they can select the instance by the QS score.
5. Construct two settings $S_{base}$ and $S_{large}$  and different selection scheme to verify the effectiveness of QaDS, try to better demonstrate the effectiveness of QaDS

**Questions To Authors:**

Refer to "Reasons to Reject".

**Reasons To Accept:**

1. Motivation is clear and reasonable to incorporate both diversity and quality. The underlying approach is also solid.
2. Experiments demonstrate improvements with QaDS selection, also better performance compared to DeepSeek, which is the base model.

**Reasons To Reject:**

1. In Table 4, althouh QaDS-$S_{large}-4.7M$ achieves the best performance $\mathbf{48.8}$, but according to Table 2, this OpenMathMix set contains all data for mathematical reasoning, only the "General" is selected by QaDS. Thus, it is confusing that the motivation is to improve mathematical reasoning, but we actually use all the corpus there.
      1. And Table 4 only shows the performance of one dataset.
      2. Although QaDS-$S_{large}-5M$ is worse, but again, we only perform QaDS on "General" according to Table 2, readers will be concerning about what's really improving here.

---

> ### Author Rebuttal · Authors · 2024-05-28
>
> Dear Reviewer oJ7q,
>
> Thank you for your thoughtful review! We are pleased to provide more information to your confusion.
>
> > Only the "General" is selected by QaDS.
>
> Firstly, we do perform selection on **both reasoning and general** data in *S_{base}*, and we achieve the best results as shown in paper "Figure 2".
>
> When coming to *S_{large}*, we aim to **give some insights on a wide range of data sources**:
>
> * **Scaling up reasoning data is helpful**: In mathematical reasoning, a common law is that mathematical reasoning consistently improves with increasing data amount [1]. And our comparison between QaDS-*S_{large}*-4.3M and 2.6M in paper "Table 4" also validates it.
> * **General data can also be influential for reasoning**: According to [1] and [2], other relevant tasks can also be helpful for reasoning. So we contain general data in our setting, and successfully selects the helpful general data by QaDS.
>
> Therefore, we contain **all reasoning data and a part of influential general data selected by QaDS** as our OpenMathMix.
>
> > Concerning about what's really improving.
>
> Our improvement comes from two parts:
>
> * The direct improving mathematical ability by more mathematical reasoning data.
> * The indirect improving reasoning-related ability by influential general data. We show cases in paper "Appendix Table 9 and 10".
>
> > Table 4 only shows the performance of one dataset.
>
> We provide our reason in paper "Section Evaluation Datasets": we adopt a subset of **MATH as *D_{t}* when computing quality scores** in *S_{large}* , and thus choose the **corresponding test set of** **MATH for evaluation**. Also, taking into account your careful comment, we evaluate on other 6 tasks and achieve best results:
>
> |         Model         | Overall  |   MATH   |   Num.   |  Math.   |   Sim.   |   SAT.   |  MMLU.   |
> | :-------------------: | :------: | :------: | :------: | :------: | :------: | :------: | :------: |
> |    InternLM2-Math     |   35.5   |   34.6   |   43.2   |   19.9   |   25.5   |   49.5   |   40.5   |
> | DeepSeekMath-Instruct |   53.1   |   46.8   |   57.7   |   42.0   |   43.6   |   67.7   |   60.5   |
> | **QaDS-OpenMathMix**  | **55.6** | **48.8** | **67.3** | **42.2** | **46.3** | **68.2** | **60.7** |
>
> [1] How abilities in large language models are affected by supervised fine-tuning data composition. *arXiv preprint arXiv:2310.05492* (2023).
>
> [2] Specialist or generalist? Instruction tuning for specific NLP tasks. *arXiv preprint arXiv:2310.15326* (2023).

---

> > ### Comment · Reviewer_oJ7q · 2024-06-05
> > **Response to authors**
> >
> > > General data can also be influential for reasoning: According to [1] and [2], other relevant tasks can also be helpful for reasoning. So we contain general data in our setting, and successfully selects the helpful general data by QaDS.
> >
> > But again, am I right that, QaDS-S_{large}-4.3M  (48.8%)> QaDS-S_{large}-3M (45.2% ) (Table 4), here the only difference is that if the math data is selected by QaDS. But turns out using QaDS in this case is worse.
> >
> > AND, in the motivation, the goal seems to select data for mathematical reasoning.
> > (Abstract below)
> > > To go further, there exist two open questions for mathematical reasoning: how to select influential data and what is an influential data composition. For the former one, we propose a Quality-aware Diverse Selection (QaDS) strategy adaptable for mathematical reasoning.
> >
> > Thus, I also expect QaDS to select influential data in math data, rather than we simply use all of them.
> >
> >
> > 2. Thanks for the experiments on other datasets.

---

> > > ### Author Response · Authors · 2024-06-05
> > > **Author Follow-up to Reviewer oJ7q**
> > >
> > > Dear Reviewer oJ7q,
> > >
> > > Thank you for your follow-up questions.
> > >
> > > > In the motivation, the goal seems to select data for mathematical reasoning.
> > >
> > > We'd like to emphasize our goal: exploiting influential data on **a wide range** of data sources (**both reasoning and general data**) for mathematical reasoning. We will make a clearer statement in our future updates.
> > >
> > > > Here the only difference is that if the math data is selected by QaDS. But turns out using QaDS in this case is worse.
> > >
> > > We share ideas from two aspects to your concern:
> > >
> > > * As we mentioned in our previous response, there exists a common law that **mathematical reasoning consistently improves with increasing reasoning data amount** [1]. There is a **large** decrease of ~1.7M reasoning data from QaDS-*S_{large}*-4.3M to QaDS-*S_{large}*-2.6M, and thus it is reasonable to see a performance drop (from 44.2% to 42.6%), which is not caused by our strategy. However, with a decrease of ~0.3M general data from QaDS-*S_{large}*-5.0M to QaDS-*S_{large}*-4.7M, our QaDS successfully selects influential general data and achieves performance gains (from 45.6% to 48.8%), showing the effectiveness of our QaDS.
> > >
> > > **Kindly Note**: We show paper "Table 2" below for your convenience.
> > >
> > > |   Task    | *S_{large}*-4.3M | *S_{large}*-5.0M | *S_{large}*-2.6M | *S_{large}*-3.0M | *S_{large}*-4.7M |
> > > | :-------: | :--------------: | :--------------: | :--------------: | :--------------: | :--------------: |
> > > | Reasoning |       All        |       All        |     Selected     |     Selected     |       All        |
> > > |  General  |        -         |       All        |        -         |     Selected     |     Selected     |
> > >
> > > * When coming to the comparison with a fixed data size, our QaDS **significantly surpasses** random strategy, and reaches an improvement of 6.6% in paper "Figure 3" (48.8% vs 42.2%) in *S_{large}*-4.7M.
> > >
> > > In summary, our QaDS can be applied on a wide range of data (**not limited on reasoning data**) and exploit most influential data. Since we are the first to explore data selection for reasoning, we hope we can provide insights to the community and will explore more strategies in our future work. We hope that our response can address your concerns. Thanks!
> > >
> > >
> > >
> > > [1] How abilities in large language models are affected by supervised fine-tuning data composition. *arXiv preprint arXiv:2310.05492* (2023).

---

> > > > ### Comment · Reviewer_oJ7q · 2024-06-05
> > > > **Response to authors**
> > > >
> > > > Got it. I think I get your point, I will raise the score.

---

### Official Review · Reviewer_rr87 · 2024-05-13

**Rating:** 6
**Confidence:** 3
**Ethics Flag:** 1

**Summary:**

This paper aims to propose the data curation method which identifies highly influential data within a large-scale mathematical reasoning dataset.

### Quality

The paper is generally clear and effectively communicates the problem it addresses. However, it lacks a crucial discussion on the computational costs involved in curating the dataset using the proposed method.

### Clarity

The paper is well-written, but some sections may be challenging to grasp on first reading.

### Originality

The application of the K-center Greedy method is not particularly novel, as it adapts an existing approach to a new problem. Nonetheless, the introduction of a quality scoring system using a zero-shot and one-shot inference appears to be a novel contribution.

### Significance

The approach to curating the training dataset is significant. This paper addresses the important issue of fine-tuning language models for reasoning datasets.

**Reasons To Accept:**

- **Thorough analysis:** This paper introduces a strategy for selecting data for a mathematical reasoning dataset. It applies this method to two different dataset sizes, S_{base} and S_{large}, demonstrating that the proposed method outperforms existing baselines. Additionally, it provides insight into the relationship between estimated and actual quality scores, showing a strong correlation with the actual scores.

**Reasons To Reject:**

- **Needs clearer writing**: Some parts of the paper are difficult to follow. For example, it is not clear which dataset is referred to in Table 3. The explanation of the scoring method in Figure 1, particularly the bi-directional arrows between the one-shot and zero-shot score blocks, is also confusing.

- **Insufficient discussion on computational costs**: The paper does not adequately address the computational costs required to embed data for diversity measurement and computing zero-shot and one-shot scores using LLMs like LLaMA-2 and Mistral. The cost implications of these computational demands, especially when seeking a modest accuracy improvement of 2% as shown in Table 4, need further discussion.

---

> ### Author Rebuttal · Authors · 2024-05-28
>
> Dear Reviewer rr87,
>
> Thank you for your comprehensive and meticulous review! We are pleased to address your points below.
>
> > It is not clear which dataset is referred to in Table 3.
>
> We use 69K selected data from *S_{base}* in this table. A detailed composition can be found in paper "Appendix Table 6". We will add a clearer summary in future updates.
>
> > The explanation of the scoring method in Figure 1 is confusing.
>
> The bi-directional arrows indicate the comparison between one-shot scores and corresponding zero-shot scores, showing the process of paper "Equation (3)". Also, we will polish the figure and make it clearer.
>
> > The paper does not adequately address the computational costs.
>
> We take our *S_{base}*-69K setting as an example (from 428K all data to select 69K data). We will show our computational costs on LLaMA-2 (for there is no obvious gap between LLaMA-2 and Mistral) from two aspects:
>
> * Diversity costs: It takes about **4.43** gpu hours to compute 428K data embeddings. The K-center Greedy process is implemented on CPU and there are no GPU costs.
>
> * Quality costs: we only use a part of representative samples to perform one-shot inference, i.e. n1=2,000 and n2=100 in our paper "Appendix Section A". In this case, the computation of zero-shot and one-shot inference takes about **0.02** and **8.42** gpu hours per dataset. After that, we train a scorer and infer the 428K scores taking **2.00** and **3.32** gpu hours respectively.
>
> We also compute the costs of another popular strategy Deita from our implement and their paper. Deita contains the process of obtaining embeddings, training quality and complexity scorers, and inferring scores. We  show the gpu hours and overall accuracy in the table below.
>
> | Strategy | Diversity Costs | One-shot Costs | Scorer Training Costs | Scoring Costs | Total  Costs | Overall Accuracy |
> | :------: | :-------------: | :------------: | :-------------------: | :-----------: | :----------: | :--------------: |
> |  Deita   |      4.43       |       -        |        > 4.00         |     6.62      |   > 15.05    |       28.9       |
> | **QaDS** |      4.43       |      8.44      |         2.00          |     3.32      |    18.19     |       31.3       |
>
> It turns out that our costs are close to Deita. Moreover, once we finish the training of our scorer, the most expensive process, i.e. the one-shot inference, is not used. We can perform fast scoring on different reasoning tasks fully based on our scorer.

---

### Author Response · Authors · 2024-06-04

We greatly appreciate the time you've invested in reviewing our paper! Having submitted our rebuttal, we are eager to know if you have any remaining concerns or require further clarification. Since the discussion phase will end soon (June 6, 2024), we would greatly appreciate your support and valuable feedback! We are more than happy to answer any further questions you may have.

Best,

Submission310 Authors

---

### Decision · Program_Chairs · 2024-07-10

**Decision:**

Accept

**Comment:**

This paper explores the selection of influential data for mathematical reasoning tasks, proposing a Quality-aware Diverse Selection (QaDS) strategy and constructing OpenMathMix, a mixture of influential data selected by QaDS.

Strengths:
1. The quality scoring is novel and could contribute to the field of data curation for mathematical reasoning (Reviewer rr87, Reviewer oJ7q).
2. The paper is well motivated -- incorporating both diversity and quality in data selection is important for reasoning tasks (Reviewer rr87, Reviewer oJ7q).

Weaknesses:
1. Generalizability (or task uniqueness): It is unclear how the method addresses any specific challenges in math reasoning, as the challenge and the approach seems quite sharable with various more general tasks (Reviewer gRx8). Concurrently, Reviewer YKSQ asked whether the approach could be "applied to other tasks than the mathematical reasoning"
2. Marginal Improvements: The reported improvements in results does not seem too significant (Reviewer gRx8).
3. Reviewers also asked for more discussions on computational costs in curating the dataset using the proposed method (Reviewer rr87).

The authors provided additional experiments on general tasks, and provided more discussions on the method efficiency and writing clarity. Unfortunately, in a reviewer discussion thread, gRx8 mentioned that they still felt that "there is no strong motivation and findings for math reasoning". That said, given that the method can generalize to more non-math tasks, I think the paper could still be beneficial if the authors could explicitly discuss the generalizability potential in a revised paper.